# Clinical Characteristics and Survival of Ovarian Cancer Patients According to Homologous Recombination Deficiency Status

**DOI:** 10.3390/cancers17101628

**Published:** 2025-05-12

**Authors:** Yagmur Sisman, Lone Schejbel, Tine Henrichsen Schnack, Claus Høgdall, Estrid Høgdall

**Affiliations:** 1Department of Gynecology, Copenhagen University Hospital, Rigshospitalet, 2100 Copenhagen, Denmark; yagmur.sisman@regionh.dk (Y.S.); claus.hogdall@regionh.dk (C.H.); 2Department of Pathology, Copenhagen University Hospital, Herlev University Hospital, 2730 Herlev, Denmark; lone.schejbel.dupont@regionh.dk; 3Department of Gynecology, Odense University Hospital, 5000 Odense, Denmark; tine.henrichsen.schnack@rsyd.dk

**Keywords:** ovarian cancer, homologous recombination deficiency, high-grade serous carcinoma, precision medicine, clinical outcomes

## Abstract

Homologous recombination deficiency (HRD) is an important biomarker in ovarian cancer, as it helps predict response to poly (ADP-ribose) polymerase (PARP) inhibitors. Nearly half of ovarian cancer patients exhibit HRD, yet the impact of HRD status on clinical outcomes in patients who are PARP inhibitor-naïve, remains unclear. This study aims to evaluate platinum-sensitive high-grade serous ovarian cancer (HGSC) patients’ clinical characteristics and survival outcomes based on their HRD status. Furthermore, we aim to explore whether platinum-resistant patients with homologous recombination repair (HRR) gene mutations are HRD-positive.

## 1. Introduction

Epithelial ovarian cancer (EOC) is the most lethal gynecologic malignancy in women, with high-grade serous ovarian cancer (HGSC) being the most common subtype [1,2]. The traditional standard of care for newly diagnosed advanced EOC is a combination of surgical cytoreduction followed by platinum-based chemotherapy. However, the therapeutic landscape has evolved with the arrival of precision medicine, introducing targeted therapies such as anti-angiogenic agents and poly (ADP-ribose) polymerase (PARP) inhibitors [3,4,5].

Homologous recombination deficiency (HRD) has emerged as a predictive biomarker in HGSC, associated with response to both PARP inhibitors and platinum-based chemotherapy [6,7,8,9]. HRD is observed in approximately 50% of HGSC cases [10]. This deficiency is attributed to germline and somatic mutations in *BRCA1/2*, epigenetic silencing of the *BRCA1* promoter via hypermethylation, and deficiencies in other homologous recombination repair mechanisms such as alterations in homologous recombination repair (HRR) genes other than mutations in *BRCA1/2* [10,11,12].

The efficacy of PARP inhibitors in platinum-resistant ovarian cancer patients has been evaluated in a few randomized clinical trials [13,14,15,16]. These studies have failed to demonstrate a significant benefit of PARP inhibition. However, none of the trials have assessed HRD status, which may be an important factor in predicting treatment response. As HRD positivity is associated with increased sensitivity to PARP inhibitors, it remains unclear whether a subset of platinum-resistant patients—specifically those with HRD-positive tumors—could still derive clinical benefit from such treatment.

This study aims to evaluate the clinical characteristics and survival outcomes of platinum-sensitive PARP-inhibitor naïve HGCS patients, stratified by HRD status. By employing two distinct HRD algorithms—an in-house genomic instability score (GIS) and the normalized large-scale state transitions score (nLST)—we investigate whether HRD-positive and HRD-negative patients differ in terms of clinical data and survival [17,18]. To the best of our knowledge, no prior studies have explored these associations in this specific patient population. Secondly, we aim to investigate whether platinum-resistant patients with HRR gene mutations are HRD-positive.

## 2. Materials and Methods

### 2.1. Patients and Tissue Samples

A total of 128 HGSC patients were selected from a previously reported cohort, on whom we had performed DNA sequencing to identify druggable targets, including potential eligibility for PARP inhibitors (Figure 1) [19]. Among these, 79 patients were classified as either platinum-sensitive (*n* = 47) or partially platinum-sensitive (*n* = 32) and were included based on their eligibility for PARP inhibitor treatment according to current national guidelines [3,4,5]. Eight of these patients were excluded due to insufficient tumor material, either because of a low tumor fraction or an inadequate amount of tissue material, leaving a final cohort of 71 chemosensitive patients for further analyses. Among the 35 platinum-resistant patients, 6 (17%) were found to carry pathogenic mutations in HRR genes and were therefore included in the study due to the presence of potentially druggable targets for PARP inhibition in *ATM*, *BRCA1* (*n* = 2), *BRCA2*, *CDK12*, and *CHEK1*. This resulted in a final cohort of 77 HGSC patients with HRD status assessed.

None of the patients in the study received treatment with PARP inhibitors. Sequencing was performed using DNA extracted from diagnostic tumor samples in a previous study from our department [19]. All patients were enrolled in the Pelvic Mass study/GOVEC study between October 2004 and October 2009. Clinical data for the patients were recorded in the Danish Gynecological Cancer Database (DGCD) [20]. Platinum-sensitivity was defined as the absence of relapse or progressive disease within 12 months after completing first-line platinum-based chemotherapy. Partial platinum-sensitivity was defined as relapse occurring between 6 and 12 months after treatment completion, whereas platinum-resistance was defined as relapse or progressive disease within six months after the end of chemotherapy. The diagnosis of HGSC was confirmed by reviewing the original tissue samples by an experienced gynecologic oncology pathologist [19].

### 2.2. DNA Extraction, Library Preparation, and Sequencing

DNA extraction, library preparation, and sequencing were performed as previously described [19]. FFPE blocks containing tumor-rich areas were selected by a gynecologic oncology pathologist, and genomic DNA was extracted using the Maxwell^®^ RSC DNA FFPE Kit (Promega, Madison, WI, USA). DNA concentration was measured with the QubitTM dsDNA High-Sensitivity Assay kit (Thermo Fisher Scientific, Waltham, MA, USA). Library preparation for the OCAv3 panel, covering 161 cancer-related genes including druggable targets, was performed manually. Sequencing was performed on the Ion S5™ XL Sequencer (Thermo Fisher Scientific, Waltham, MA, USA), and data were analyzed using Ion Reporter™ Software (v. 5.14) and the Oncomine Comprehensive v3–w4.0–DNA–Single Sample workflow. Mutational data analysis was performed using Python (v. 3.7), as previously described [19,21,22]. Only pathogenic and likely pathogenic variants, defined by ClinVar and classified according to ACMG criteria, were included [23]. This study exclusively included somatic mutation analyses. Germline testing was not performed, and potential germline variants were therefore not identified.

### 2.3. HRD Analysis

All tumor samples were analyzed for HRD using the OncoScan CNV FFPE Assay, as previously described [17]. HRD assessment could not be performed in 8 out of the 77 platinum-sensitive patients due to insufficient tumor material. For the remaining samples where HRD analysis was carried out, the calculated allele-specific copy number–based tumor fraction ranged from 0.27 to 1. Twenty-one patients had previously had their HRD status measured as part of a validation study in our department [17]. HRD assessment was performed with two published R-based algorithms, generating an in-house GIS and nLST [17,18,24]. Genomic instability was evaluated by calculating loss of heterozygosity (LOH), large-scale state transitions (LST), and telomeric allelic imbalance (TAI) scores. The in-house GIS was calculated as the sum of these three parameters, with a local cut-off of in-house GIS of ≥50 used to define HRD positivity [17]. For nLST analysis, the OncoScanR v1.6.0 package was used, and HRD-positivity was determined by a cut-off of nLST ≥ 15 [18]. The cut-off value for in-house GIS (≥50) was based on a previously published validation study, in which the in-house GIS assay was compared with the Myriad myChoice^®^ HRD test. Using receiver operating characteristic (ROC) curve analysis, an optimal cut-off of 49.75 was identified, providing a diagnostic accuracy of 85%, a low false-positive rate (12.3%), and a false-negative rate of 1.7%. The area under the curve (AUC) was 0.968, indicating excellent test performance. For consistency and ease of interpretation, the cut-off was rounded to 50 and applied in this study [17]. A similar approach was used for the nLST score, with a threshold of ≥15 derived from validation against the PAOLA-1 cohort [18]. It is important to note that the in-house GIS and nLST represent two distinct algorithms, each with different methodologies and cut-offs. Specifically, nLST is not a direct measure of the LST component used in the GIS, but rather a separate metric developed independently to assess genomic instability.

### 2.4. Statistical Analysis

All statistical analyses were performed using Python (v. 3.7) and R [19]. The Shapiro–Wilk test was used to assess the normality of the data, which indicated that the data were not normally distributed. The Wilcoxon Rank-Sum Test was employed to determine whether there were significant differences between the groups for the various continuous variables analyzed. For categorical variables, the Chi-squared test was used when expected cell counts were sufficiently large (≥5); otherwise, Fisher’s exact test was applied. Survival outcomes between groups were compared using Kaplan–Meier analysis and log-rank test.

## 3. Results

### 3.1. Patients

HRD status was successfully assessed in 77 individuals included in this study. A total of 40 patients were platinum-sensitive, 31 patients were partially platinum-sensitive and 6 patients were platinum-resistant (Figure 1). Seventy-five patients were treated with carboplatin and taxol and two patients were treated with carboplatin alone. Seventy-five patients received six cycles of carboplatin and taxol, one received five cycles with carboplatin and one received four cycles with carboplatin and taxol. No differences in clinical characteristics and chemotherapy treatment between included and excluded patients were observed.

### 3.2. HRD Analysis in Platinum-Sensitive Patients

Using the in-house GIS, 37 patients (52%) were classified as HRD-positive, while 34 (48%) were HRD-negative. Applying the nLST algorithm, 43 patients (61%) were identified as HRD-positive, and 28 (39%) as HRD-negative (Figure 1). Twelve patients (17%) displayed discordant HRD status between the two algorithms, the majority of those being positive with the nLST algorithm and negative with the in-house GIS-score (Figure 2). No significant differences in in-house GIS (*p* = 0.91) or nLST (*p* = 0.59) scores were observed between platinum-sensitive and partially platinum-sensitive patients (Figure 3 and Figure 4). Similarly, when comparing *BRCA1*/2-mutated patients with HRD-positive *BRCA1*/2-wt and HRD-negative patients, no significant differences in in-house GIS (*p* = 0.98) or nLST (*p* = 0.33) scores were found.

### 3.3. Clinical Characteristics and Survival in Platinum-Sensitive Patients

We compared clinical characteristics between HRD-positive and HRD-negative platinum-sensitive HGSC patients, including age, performance score, disease stage, and surgical radicality (Table 1 and Table 2). No significant differences were observed between HRD-positive and HRD-negative patients, whether defined by the in-house GIS or nLST score. Furthermore, no significant clinical differences were identified between HRD-positive patients defined by the in-house GIS and nLST algorithms, nor were there differences among HRD-negative patients across these classifications. Similarly, no differences in overall survival were found between HRD-positive and HRD-negative patients, regardless of the algorithm used (in-house GIS: *p* = 0.79; nLST: *p* = 0.76) (Appendix A). Only patients with FIGO stage 3 and 4 diseases were included in the survival analysis.

A total of 12 patients demonstrated discordant HRD status depending on whether the in-house GIS or nLST algorithm was applied (Figure 2). Of these, nine (75%) patients were classified as negative according to the in-house GIS. Among these, two patients had scores near the cutoff, with GIS values of 48 and 49, respectively. Three patients were positive according to the in-house GIS but negative according to the nLST algorithm. Two of these patients scored close to the cutoff (nLST scores of 13.5 and 14.5, respectively; Figure 2). The 12 patients with discordant HRD status were compared to the 59 patients with concordant HRD status to identify any differences in clinical characteristics, including age, performance status, or tumor stage. Statistical analysis revealed no differences between the discordant and concordant groups, although the small number of patients in the discordant group limits the ability to draw firm conclusions (Appendix A).

The only significant clinical differences observed were in age; patients with *BRCA1*/2 mutations in the tumor tissue were younger compared to *BRCA1*/*2* wt patients (*p* = 0.001). This age difference remained significant when comparing *BRCA1*/2 mutated patients to HRD-positive *BRCA1*/2 wt patients, whether defined by the in-house GIS (*p* = 0.006) or nLST score (*p* = 0.004). Using the nLST algorithm, a significant difference in CA125 levels was observed between HRD-positive and HRD-negative patients (*p* = 0.014) (Table 2). However, this significance was driven by a single outlier with an exceptionally high CA125 level of 17,160. When this observation was excluded from the analysis, the difference was no longer statistically significant. No significant difference in survival was observed between platinum-sensitive patients with *BRCA1/2* mutations and those with BRCAwt (*p* = 0.39) (Appendix A).

### 3.4. Mutational Analysis

Among the included 71 platinum-sensitive patients, 65 (92%) had 1 or more pathogenic or likely pathogenic mutations. These mutations were distributed across 29 cancer-related genes, including *TP53* mutations in 63 patients (89%) and *BRCA1*/2 mutations in 16 patients (23%) (Appendix A). Only the genes *TP53* (89%), *BRCA1* (18%), and *CREBBP* (8%) were mutated in more than 5% of the patients. Mutations in *CREBBP* were identified in six patients, five of whom were classified as HRD-positive. In the HRD-positive group defined by in-house GIS, 34 patients (92%) had *TP53* mutations, compared to 29 patients (85%) in the HRD-negative group. Similarly, when HRD status was defined by nLST score, 39 HRD-positive patients (91%) and 24 HRD-negative patients (83%) had *TP53* mutations. Eighteen platinum-sensitive patients (49%) had mutations in HRR genes, including *BRCA1*/2 (*n* = 16), *ATM* (*n* = 1), and *CDK12* (*n* = 1). All of them were HRD positive with both algorithms. Thus, 20 HRD-positive patients without HRR gene mutations were identified using the in-house GIS, while 26 were identified using the nLST score.

### 3.5. Platinum-Resistance and HRD

HRD analyses were conducted in six platinum-resistant HGSC patients harboring HRR gene mutations, including *BRCA1* (*n* = 2), *BRCA2*, *ATM*, *CHEK1*, and *CDK12* (Appendix A). Among these, four patients (67%) were HRD-positive, while two patients (33%) were HRD-negative. HRD status was consistent regardless of the used algorithm (Appendix A). The HRD-positive patients had pathogenic mutations in *BRCA1*, *BRCA2,* and *CDK12*, while the HRD-negative patients had pathogenic mutations in *ATM* and *CHEK1*. All platinum-resistant patients completed six full doses of chemotherapy. Additionally, we analyzed the timing of relapse among the four HRD-positive platinum-resistant patients. The relapses did not cluster around the 6-month cut-off.

## 4. Discussion

Clinical characteristics and survival outcomes in PARP inhibitor-naïve platinum-sensitive HGSC patients, using both the in-house GIS and nLST scores, revealed no significant differences between HRD-positive and HRD-negative groups, except that *BRCA1*/2 mutated patients were younger. Therefore, our findings are novel and provide valuable insights into this patient population.

Interestingly, one study has examined the association between HRD status and the impact of different cancer spread patterns (miliary versus non-miliary) in a PARP inhibitor-naïve cohort [25]. It was observed that tumors exhibiting miliary spread were exclusively HRD-negative, and patients with these tumors demonstrated significantly poorer survival outcomes compared to those with HRD-negative tumors exhibiting non-miliary spread. Notably, patients with HRD-negative tumors with non-miliary spread exhibited survival outcomes comparable to those with HRD-positive tumors [25]. This finding highlights the possible heterogeneity among HRD-negative patients and reinforces the need to assess specific tumor characteristics, such as spread patterns, to enhance our understanding of survival outcomes.

In a previous study, we identified that HGSC patients with somatic *BRCA1/2* mutations had a longer overall survival compared to *BRCAwt* patients [19]. In this study, including platinum-sensitive patients only, no survival benefit was observed in patients with *BRCA1/2* mutations compared to those without such mutations. Similarly, when analyzing HRD status within this platinum-sensitive PARP inhibitor-naïve population, we found no significant survival advantage for HRD-positive patients independent of the algorithms used. This suggests that platinum sensitivity itself is a stronger independent prognostic factor of survival outcomes than HRD status.

In our study, platinum sensitivity is defined as the absence of relapse or progressive disease, or a relapse occurring more than six months after completing first-line chemotherapy. Relapse and progression were determined using the best clinical evaluations based on imaging modalities such as CT, MRI, and PET-CT scans, along with serum CA125 levels and patient symptoms. However, the definition of platinum sensitivity remains a topic of ongoing debate due to the significant variability in its criteria across studies. As highlighted in a newly published review, platinum sensitivity can range from the absence of relapse within six months to progression-free intervals exceeding 12 months [26,27]. This variability complicates cross-study comparisons and may influence how patients are selected for PARP inhibitor treatment.

Furthermore, we applied two algorithms for HRD status. While the in-house GIS algorithm has been validated against Myriad, the nLST score has undergone clinical validation against the PAOLA cohort [17,18]. In the clinical validation, the nLST test identified more HRD-positive cases among *BRCA*wt patients compared to the Myriad test [18]. This difference is mainly explained by the threshold used; the nLST cut-off of 15 corresponds to a Myriad GIS of 38. Patients falling into the “HRD low positive” group (15 ≤ nLST < 20) responded better to PARP inhibitor treatment than HRD-negative patients, with a 1-year PFS similar to HRD high positive (nLST > 20) but with a lower 2-year PFS [18]. In our study, we found more HRD-positive patients using the nLST score, as expected. This was the case both when using a cut-off of 15, and when applying a higher cut-off of 18, corresponding to the Myriad threshold. The study presents that the choice of the algorithm significantly impacts the proportion of patients identified as HRD-positive, directly influencing treatment eligibility. Previous studies have also highlighted this challenge, which remains a key barrier to optimizing patient outcomes [28,29,30]. This underscores the importance of thoroughly validating the algorithms and assays used to define the HRD status in a clinical setting.

Twelve patients showed discordant HRD status between the GIS and nLST algorithms. We did not observe any clear clinical patterns distinguishing discordant patients from those with concordant HRD results, such as differences in age, FIGO stage, platinum sensitivity, or survival outcomes (Appendix A). However, the small sample size of the discordant group limits the strength of the conclusions. One possible explanation for the discordance is that these patients harbor intermediate levels of genomic instability that place them close to the scoring thresholds. These findings suggest that discordant patients might represent an intermediate HRD phenotype, highlighting the complexity of accurately classifying HRD and the need for further refinement of HRD assays.

*BRCA1/2* mutated patients were younger than *BRCA1*/*2* wt patients, likely due to genetic predisposition. However, we did not perform germline analyses to confirm hereditary mutations. Additionally, we observed that three platinum-resistant patients harbored *BRCA1/2* mutations. None of the specific *BRCA1/2* mutations could explain why these patients were platinum-resistant. All three were HRD-positive according to both algorithms, suggesting a biallelic *BRCA1*/*2* defect in these cases. Our in-house GIS was not close to the cut-off for any of these patients. Nevertheless, it has previously been reported that HRD-positive patients can still exhibit platinum resistance, likely due to the activation of alternative resistance mechanisms [31].

Few randomized studies have investigated the impact of PARP inhibitors on PFS in platinum-resistant ovarian cancer patients [13,14,15,16]. None of these trials have demonstrated a significant benefit, and importantly, none have determined HRD status in the included populations. However, using archival tumor samples, the OCTOVA trial plans further biomarker analyses to assess tumor HRD status, including *BRCA1/2* mutation status. Furthermore, prior studies investigating PARP inhibitors in women with relapsed platinum-resistant ovarian cancer have shown that treatment response is largely influenced by *BRCA1/2* mutation status. Patients harboring *BRCA1/2* mutations have demonstrated response rates of up to 30% with olaparib, whereas significantly lower response rates—ranging from 3% to 13%—have been reported in *BRCA1/2* wt patients, as shown in trials such as QUADRA (niraparib) and CLIO (olaparib) [32,33,34]. In our cohort, we identified four HRD-positive patients among the platinum-resistant subgroup of six patients with HRR gene mutations, raising the question of whether a subgroup of patients within this typically treatment-resistant population could still benefit from PARP inhibitor therapy. Future studies focusing specifically on HRD-positive platinum-resistant patients are therefore warranted.

Tumor heterogeneity can be a challenge in assessing HRD status as it may vary depending on which tumor biopsy is analyzed, even when samples are taken from the same patient on the same day. A study by Sztupinszki et al. demonstrated that biopsies from different anatomical locations can yield divergent HRD results [35]. In an SNP array-based cohort, individual patients had up to 27 biopsies taken from various locations or at different time points [36]. In 8 out of the 16 cases, the HRD score varied significantly, with values fluctuating above and below the accepted HRD threshold of 42 [37]. They also extracted three DNA samples from the same primary surgery material, and conducted whole-exome sequencing, and found that the HRD scores were remarkably consistent [38]. However, none of the studies report the tumor fraction in the tissue samples, making it difficult to evaluate the quality of the HRD analyses. In our study, HRD status was assessed based on a single tissue sample per patient. This is a common approach and is supported by the study by Wahlde et al., who collected three biopsies from the same breast tumor in 33 patients. HRD status was assessed in 70 of the biopsies, and they found that HRD scores were highly consistent between different biopsies obtained from the same tumor [39].

The present study demonstrates that HRD status, as determined by both in-house GIS and nLST algorithms, is not associated with distinct clinical characteristics or survival in platinum-sensitive, PARP inhibitor-naïve HGSC patients. These findings suggest that, within this specific clinical context, HRD testing may have limited prognostic value. Nevertheless, it is important to emphasize that our results do not undermine the established role of HRD as a predictive biomarker for response to PARP inhibitors. Rather, they underscore the need to clearly distinguish between the predictive and prognostic applications of HRD testing. While our data do not support its use as a prognostic tool in PARP inhibitor-naïve settings, HRD testing remains clinically relevant for guiding therapeutic decisions, particularly in identifying patients likely to benefit from PARP inhibitor treatment [3].

A key strength of this study is the inclusion of patients outside the context of randomized clinical trials, allowing for a more representative population unrestricted by strict eligibility criteria. This approach represents a broad cross-section of the population, reducing selection bias. Unlike clinical trials with strict criteria, this study captures the heterogeneity of patients typically encountered in routine practice. By utilizing data from a real-world cohort, we address these limitations and provide a more accurate representation of clinical characteristics and survival outcomes among PARP inhibitor-naïve platinum-sensitive HGSC patients [40]. Another strength of our study is the use of two distinct algorithms to define HRD status: the in-house GIS and the nLST score. Notably, the consistent results produced by both algorithms reinforce the validity of our findings [41].

Lastly, one limitation of our study is that some of our analyses, such as those focusing on mutational analysis and platinum-resistant patients, were descriptive in nature due to limited statistical power caused by the small number of patients or mutations [42]. In particular, the subgroup of platinum-resistant patients with HRR mutations was extremely small (*n* = 6), with only four patients being HRD-positive. This limits the generalizability and robustness of any conclusions regarding the potential benefit of PARP inhibitors in this group. As such, the subgroup analyses’ results should be interpreted cautiously and viewed as hypothesis-generating rather than conclusive. Larger studies are needed to validate these preliminary observations.

## 5. Conclusions

Our findings suggest that HRD status is not associated with clinical characteristics or survival outcomes in PARP inhibitor-naïve platinum-sensitive HGSC patients. These results emphasize the need for further research to explore other potential markers or factors influencing clinical outcomes. Interestingly, we identified four HRD-positive patients among six platinum-resistant cases with HRR gene mutations, highlighting a potential subgroup that may benefit from PARP inhibitor therapy. Therefore, future studies focusing on HRD-positive patients within the platinum-resistant population are warranted.

## Figures and Tables

**Figure 1 cancers-17-01628-f001:**
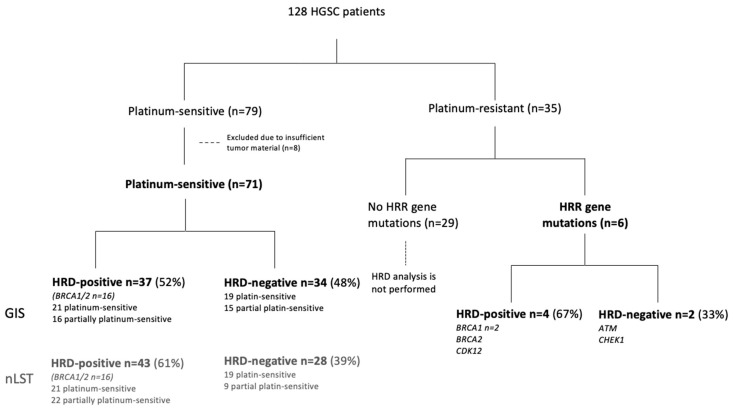
Flow diagram of patient inclusion and HRD status. GIS, genomic instability score; HGSC, high-grade serous ovarian cancer; HRD, homologous recombination deficiency; HRR, homologous recombination repair; nLST, normalized large-scale transition score.

**Figure 2 cancers-17-01628-f002:**
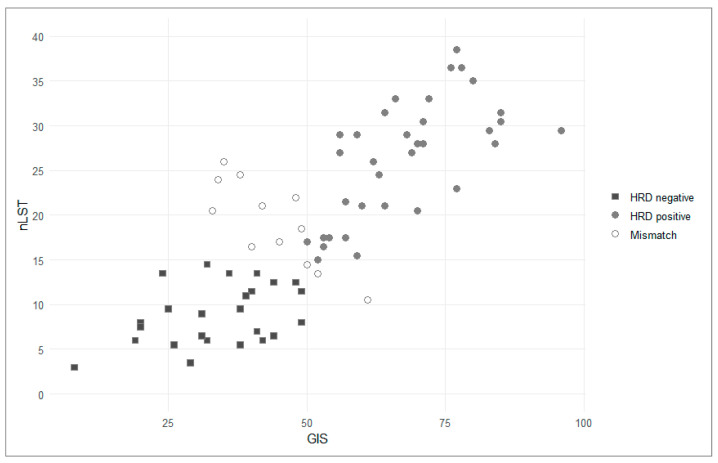
HRD status according to in-house GIS and nLST. Mismatch refers to discordant HRD status between two algorithms. GIS, genomic instability score; HRD, homologous recombination deficiency; nLST, normalized large-scale transition score.

**Figure 3 cancers-17-01628-f003:**
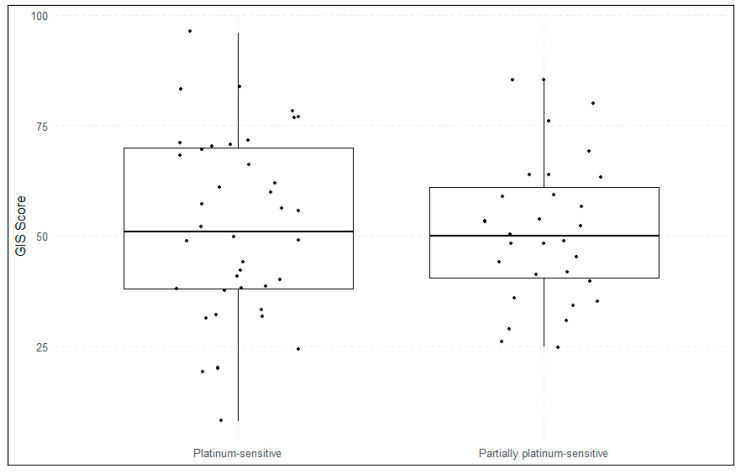
In-house GIS according to platinum-sensitivity. The horizontal line in the middle of each box represents the median. GIS, genomic instability score.

**Figure 4 cancers-17-01628-f004:**
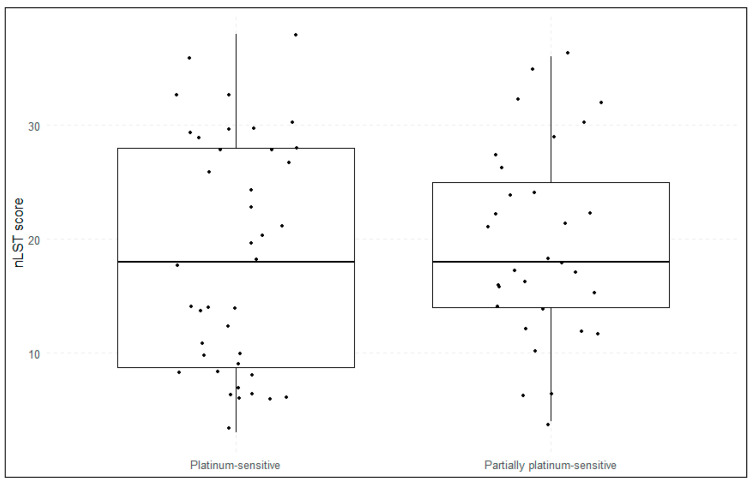
nLST according to platinum-sensitivity. The horizontal line in the middle of each box represents the median. nLST, normalized large-scale transition score.

**Table 1 cancers-17-01628-t001:** Clinical characteristics of the platinum-sensitive HGSC patients according to HRD status measured with in-house GIS.

	HRD-Positive (N = 37)	HRD-Negative (N = 34)	*p*-Value
Age in years median (range)	59 (41–89)	66 (42–87)	0.057
CA125 median (range)	728 (46–17,160)	416 (30–7230)	0.053
RMI median (range)	5109 (252–51,480)	3155 (63–65,070)	0.156
BMI median (range)	24 (19–34)	24 (19–41)	0.950
Follow up time in months median (range)	87 (63–121)	87 (63–121)	0.612
**Performance score**			
0–1	35 (95%)	28 (82%)	0.756
≥2	2 (5%)	8 (18%)	1.000
**FIGO stage**			
I–II	7 (19%)	7 (21%)	0.592
III–IV	30 (81%)	27 (79%)	0.238
**Residual tumor after surgery**			
0–1	24 (65%)	25 (74%)	0.821
≥1	13 (35%)	9 (26%)	1.000
**Platinum response**			
>12 months (sensitive)	21 (57%)	19 (56%)	1.000
6–12 months (partial sensitive)	16 (43%)	15 (44%)	1.000
**Survival in months**			
OS median (range)	55 (11–174)	49 (16–189)	0.284
PFS median (range)	23 (8–107)	19 (10–95)	0.467

FIGO, International Federation of Gynecology and Obstetrics; HGSC, high-grade serous ovarian cancer; HRD, homologous recombination deficiency; GIS, genomic instability score; OS, overall survival; PFS, progression free survival; RMI, risk of malignancy index.

**Table 2 cancers-17-01628-t002:** Clinical characteristics of platinum-sensitive HGSC patients according to HRD status measured with nLST.

	HRD-Positive (N = 43)	HRD-Negative (N = 28)	*p*-Value
Age in years median (range)	62 (41–89)	67 (42–87)	0.084
CA125 median (range)	728 (46–17,160)	258 (30–7230)	0.014
RMI median (range)	7092 (252–51,480)	2885 (63–65,070)	0.147
BMI median (range)	25 (19–35)	24 (19–41)	0.269
Follow up time in months median (range)	88 (63–123)	87 (65–123)	0.517
**Performance score**			
0–1	38 (88%)	25 (89%)	0.608
≥2	5 (12%)	3 (11%)	0.375
**FIGO stage**			
I–II	9 (21%)	5 (18%)	1.000
III–IV	34 (79%)	23 (82%)	0.423
**Residual tumor after surgery**			
0–1	28 (65%)	21 (75%)	0.750
≥1	15 (35%)	7 (25%)	1.000
**Platinum response**			
≥12 months (sensitive)	21 (49%)	19 (68%)	0.182
6–12 months (partial sensitive)	22 (51%)	9 (32%)	0.182
**Survival in months**			
OS median (range)	54 (11–174)	57 (16–189)	0.757
PFS median (range)	20 (8–107)	22 (10–95)	0.580

FIGO, International Federation of Gynecology and Obstetrics; HGSC, high-grade serous ovarian cancer; HRD, homologous recombination deficiency; nLST, normalized large-scale transition score, OS, overall survival; PFS, progression free survival; RMI: risk of malignancy index.

## Data Availability

Data are contained within the article.

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
