# Peer review of "Clinical Characteristics and Survival of Ovarian Cancer Patients According to Homologous Recombination Deficiency Status"

_cancers, 2025, doi:10.3390/cancers17101628_

Round 1

Reviewer 1 Report

Comments and Suggestions for Authors

This manuscript addresses the clinical relevance of homologous recombination deficiency (HRD) in high-grade serous ovarian cancer (HGSC), focusing on platinum-sensitive and platinum-resistant patients naïve to PARP inhibitors. It utilizes two HRD scoring algorithms (in-house GIS and nLST) to stratify patients and investigate correlations with clinical characteristics and outcomes. The study is well-organized, provides a clear rationale, and adds value by exploring an understudied group — PARP-naïve patients.

The subgroup of platinum-resistant patients with HRR mutations is extremely small (n=6), with only four being HRD-positive. This limits the generalizability and robustness of any conclusions about the potential benefit of PARP inhibitors in this group. The authors should acknowledge this limitation more explicitly in the discussion and treat findings as preliminary.

Twelve patients showed discordant HRD status between GIS and nLST. While some analysis is presented, further exploration into clinical relevance or biological explanation is needed. Can discordant patients be clinically distinguished or associated with intermediate HRD phenotypes?

The choice of cut-offs for GIS (≥50) and nLST (≥15) affects HRD categorization. The authors note validation against Myriad and PAOLA cohorts, but further detail is needed to justify these thresholds. Were ROC curves or external validation applied in this study?

The manuscript is mostly well-written but would benefit from minor language polishing, particularly to avoid redundancy (e.g., repetition of “clinical characteristics and survival outcomes” in the abstract and introduction)

Author Response

This manuscript addresses the clinical relevance of homologous recombination deficiency (HRD) in
high-grade serous ovarian cancer (HGSC), focusing on platinum-sensitive and platinum-resistant
patients naïve to PARP inhibitors. It utilizes two HRD scoring algorithms (in-house GIS and nLST)
to stratify patients and investigate correlations with clinical characteristics and outcomes. The study
is well-organized, provides a clear rationale, and adds value by exploring an understudied group —
PARP-naïve patients.
We sincerely thank the reviewer for taking the time to read and evaluate our manuscript. We
appreciate the positive feedback regarding the organization, rationale, and relevance of our study,
particularly the focus on the understudied group of PARP inhibitor–naïve patients. We are grateful
for the thoughtful comments and constructive suggestions that have helped us to improve the
manuscript.

The subgroup of platinum-resistant patients with HRR mutations is extremely small (n=6), with only
four being HRD-positive. This limits the generalizability and robustness of any conclusions about the
potential benefit of PARP inhibitors in this group. The authors should acknowledge this limitation
more explicitly in the discussion and treat findings as preliminary.
We thank the reviewer for the constructive comment. We completely agree with this important point
and have now addressed it more explicitly in the Discussion by adding the following:
In particular, the subgroup of platinum-resistant patients with HRR mutations was extremely small
(n=6), with only four patients being HRD-positive. This limits the generalizability and robustness of
any conclusions regarding the potential benefit of PARP inhibitors in this group. As such, the
subgroup analyses' results should be interpreted cautiously and viewed as hypothesis-generating
rather than conclusive. Larger studies are needed to validate these preliminary observations (lines
393-399).

Twelve patients showed discordant HRD status between GIS and nLST. While some analysis is
presented, further exploration into clinical relevance or biological explanation is needed. Can
discordant patients be clinically distinguished or associated with intermediate HRD phenotypes?
We thank the reviewer for this insightful comment. In response, we have expanded the Discussion to
further address the potential clinical and biological relevance of patients with discordant HRD status
between the GIS and nLST algorithms. Specifically, we have added the following:
Twelve patients showed discordant HRD status between the GIS and nLST algorithms. We did not
observe any clear clinical patterns distinguishing discordant patients from those with concordant
HRD results, such as differences in age, FIGO stage, platinum sensitivity, or survival outcomes
(Table S1). However, the small sample size of the discordant group limits the strength of conclusions.
One possible explanation for the discordance is that these patients harbor intermediate levels of
genomic instability that place them close to the scoring thresholds. These findings suggest that
discordant patients might represent an intermediate HRD phenotype, highlighting the complexity of
accurately classifying HRD and the need for further refinement of HRD assays (lines 321-329).

The choice of cut-offs for GIS (≥50) and nLST (≥15) affects HRD categorization. The authors note
validation against Myriad and PAOLA cohorts, but further detail is needed to justify these thresholds.
Were ROC curves or external validation applied in this study?
Thank you for your insightful comment regarding the justification of the in-house GIS and nLST cutoff values. To address this, we have now included a detailed explanation in the Methods section (2.3
HRD analysis), outlining the origin and validation of the applied thresholds. The following text has
been added:
The cut-off value for in-house GIS (≥ 50) was based on a previously published validation study, in
which the in-house GIS assay was compared with the Myriad myChoice® HRD test. Using receiver
operating characteristic (ROC) curve analysis, an optimal cut-off of 49.75 was identified, providing
a diagnostic accuracy of 85%, a low false-positive rate (12.3%), and a false-negative rate of 1.7%.
The area under the curve (AUC) was 0.968, indicating excellent test performance. For consistency
and ease of interpretation, the cut-off was rounded to 50 and applied in this study [17]. A similar
approach was used for the nLST score, with the threshold of ≥ 15 derived from validation against the
PAOLA-1 cohort [18] (lines 134-142).

The manuscript is mostly well-written but would benefit from minor language polishing, particularly
to avoid redundancy (e.g., repetition of “clinical characteristics and survival outcomes” in the abstract
and introduction)
We thank the reviewer for this helpful suggestion. In response, we have carefully revised the
manuscript to improve language clarity and avoid redundancy, particularly by adjusting repeated
phrases such as “clinical characteristics and survival outcomes” in the abstract and introduction. We
believe these changes have enhanced the overall readability of the manuscript.

Reviewer 2 Report

Comments and Suggestions for Authors

Please rewrite the Abstract and include the number of cases involved in your study.

The Ethical permission is not mentioned in the submission.

What was the initial tissue quantity during the DNA isolation?

What was the reason for the cases where you could not isolate DNA?

Please improve the quality of your figures.

Practically, it is a negative study, please explain it in more details, that do you recommend or not to perform HRD  test as biomarker, or not. It seems it does not give any useful information to the clinicians.

Author Response

Please rewrite the Abstract and include the number of cases involved in your study.
Thank you for your valuable feedback. We have revised the Abstract to improve clarity and flow and
have now included the number of cases analyzed in the study, as requested. Specifically, we state the
number of platinum-sensitive and platinum-resistant patients, as well as their HRD classification
according to both algorithms used.

The Ethical permission is not mentioned in the submission.
Thank you for the comment. The information regarding ethical approval is provided at the end of the
manuscript, under the section Institutional Review Board Statement, stating:
The study was conducted according to the guidelines of the Declaration of Helsinki, including written
informed consent from all subjects. The Danish National Committee has approved the study for
research ethics, Capital Region (approval codes; KF01-227/03 and KF01-143/04, H15020061). All
performed molecular analyses are somatic, not germline (lines 428-431).
However, if preferred, we are happy to include a separate subsection on ethical approval in the
Methods section to ensure greater visibility and clarity.

What was the initial tissue quantity during the DNA isolation? What was the reason for the cases
where you could not isolate DNA?
Thank you for your relevant questions. We have now addressed this in the Methods section 2.3 about
HRD analyses, where we have added the following clarification:
HRD assessment could not be performed in 8 out of the 77 platinum-sensitive patients due to
insufficient tumor material. For the remaining samples where HRD analysis was carried out, the
calculated allele-specific copy number–based tumor fraction ranged from 0.27 to 1 (lines 123-126).
Please let us know if further detail is required.

Please improve the quality of your figures.
Thank you for your comment. We have carefully reviewed and optimized all figures to the best of
our ability, to ensure clarity and readability. All images have been re-exported in high resolution and
checked for consistency with journal requirements. Should the reviewer or editors have any specific
suggestions or preferred formats, we would be happy to make further adjustments accordingly.

Practically, it is a negative study, please explain it in more details, that do you recommend or not to
perform HRD test as biomarker, or not. It seems it does not give any useful information to the
clinicians.
Thank you for this important comment. We agree that our findings show no significant association
between HRD status and clinical characteristics or survival outcomes in PARP inhibitor-naïve,
platinum-sensitive HGSC patients—thus, from this perspective, the study is indeed negative.
However, we believe these results provide valuable insights, especially in clarifying the limited
prognostic value of HRD testing in this specific clinical setting.
Importantly, our findings do not argue against the use of HRD as a predictive biomarker for PARP
inhibitor response, where its clinical utility is well established. Rather, our results suggest that HRD
testing may have limited additional value in informing about clinical characteristics and survival in
PARPi-naïve platinum-sensitive HGSC patients. We have clarified this interpretation in the
Discussion and added a statement on the clinical implications of our findings, emphasizing that HRD
testing remains relevant as a predictive biomarker, but not as a standalone prognostic tool in PARPnaïve platinum-sensitive populations:
The present study demonstrates that HRD status, as determined by both in-house GIS and nLST
algorithms, is not associated with distinct clinical characteristics or survival in platinum-sensitive,
PARP inhibitor-naïve HGSC patients. These findings suggest that, within this specific clinical
context, HRD testing may have limited prognostic value. Nevertheless, it is important to emphasize
that our results do not undermine the established role of HRD as a predictive biomarker for response
to PARP inhibitors. Rather, they underscore the need to clearly distinguish between the predictive
and prognostic applications of HRD testing. While our data do not support its use as a prognostic
tool in PARP inhibitor-naïve settings, HRD testing remains clinically relevant for guiding therapeutic
decisions, particularly in identifying patients likely to benefit from PARP inhibitor treatment (lines
370-380). 

Round 2

Reviewer 2 Report

Comments and Suggestions for Authors

The quality of the ms was improved, thank you for your response.